# Investigating the Effectiveness of Very Low-Calorie Diets and Low-Fat Vegan Diets on Weight and Glycemic Markers in Type 2 Diabetes Mellitus: A Systematic Review and Meta-Analysis

**DOI:** 10.3390/nu14224870

**Published:** 2022-11-17

**Authors:** Anjali Kashyap, Alexander Mackay, Ben Carter, Claire L. Fyfe, Alexandra M. Johnstone, Phyo K. Myint

**Affiliations:** 1School of Medicine, Medical Sciences & Nutrition, University of Aberdeen, Foresterhill Health Campus, Aberdeen AB25 2ZD, UK; 2Department of Biostatistics and Health Informatics, Institute of Psychiatry, Psychology and Neuroscience, King’s College London, London WC2R 2LS, UK; 3The Rowett Institute, University of Aberdeen, Foresterhill Health Campus, Aberdeen AB25 2ZD, UK; 4Institute of Applied Health Sciences, University of Aberdeen, Polwarth Building, Room 4.013, Foresterhill Health Campus, Aberdeen AB25 2ZD, UK; 5Department of Medicine for the Elderly, NHS Grampian, Cornhill Road, Aberdeen AB25 2ZN, UK

**Keywords:** diabetes mellitus, type 2, plant-based diets, very low-calorie diet, low-fat, vegan diet, weight loss, glycemic control

## Abstract

Caloric restriction and vegan diets have demonstrated protective effects for diabetes, however their role in improving clinically relevant outcomes has not been summarized. Our aim was to evaluate the evidence for low-calorie diets (VLCD) and vegan diets on weight and glycemic control in the management of patients with Type 2 Diabetes. Database searches were conducted using Cochrane Library, MEDLINE (Ovid) and Embase. Systematic Review Registration: CRD42022310299. Methodological quality of studies was assessed using Cochrane RoB Tool for RCTs, Cochrane ROBINS-I RoB Tool for non-RCTs and NIH Quality Assessment tool for other studies. Sixteen studies with a total of 834 individuals were included and assessed to have a moderate to high risk of bias. Statistically significant changes in weight, BMI, and HbA1c were not observed in vegan diet cohorts. However, LDL cholesterol was significantly decreased by vegan diet. VLCDs significantly improved glycaemic control, with reductions in fasting glucose, pooled mean difference (MD) −1.51 mmol/L (95% CI −2.89, −0.13; *p* = 0.03; 2 studies) and HbA1c, pooled MD −0.66% (95% CI −1.28, −0.03; *p* = 0.04; 3 studies) compared to non-dietary therapy. Both diets suggested a trend towards improved weight loss and anthropometric markers vs. control. VLCD diet intervention is associated with improvement in glycaemia control in patients with Type 2 Diabetes.

## 1. Introduction

Type 2 Diabetes Mellitus (T2DM) is a global epidemic, driven by an increased prevalence of obesity in both children and adults [1]. Increased consumption of calorific foods including processed foods and beverages, meat and other animal products, sugary beverages, and refined grains are believed to play a key role in the growing rates of T2DM worldwide [2]. The International Diabetes Federation estimate that approximately 537 million adults (20–79 years) are living with diabetes. The total number of people living with diabetes is projected to rise to 643 million by 2030 and 783 million by 2045 [3], with the predominant type being T2DM. With fewer than 2% of people with T2DM entering a state of spontaneous remission [4], the present clinical paradigm is that T2DM is an irreversible condition.

The primary approach for management of T2DM is to achieve and maintain weight loss [5]. Evidence has shown a multifactorial intervention in the form of caloric restriction, exercise, and behavior change has optimal effects in improving glycemic control and weight [6,7]. Current dietary interventions for the management of T2DM include restricting carbohydrates, cholesterol, and fat intake, as well as caloric restriction [8]. Dietary restriction therapies for management and cessation of T2DM have mainly focused on weight loss through the implementation of either low-calorie diets (LCD) defined as 1200–1500 kcal/day [9,10,11] or very low-calorie diets (VLCD) ranging from 450–800 kcals/day [12]. Current literature supports VLCD diets, illustrating that such diets are superior in inducing/promoting rapid weight reductions, improving insulin secretion, and lowering hemoglobin A1c (HbA1c) to levels seen in pre-diabetes or normoglycemia [13,14,15,16,17,18,19,20,21,22].

Plant-based diets, focusing on inclusion of foods from plant sources and exclusion of animal-based products, have gained recognition in public heath for not only their potential in promoting sustainability, but also to curb the onset and assist in management of chronic disease [23], including cardiovascular disease, some cancers, and T2DM [24,25,26,27]. Clinical studies have demonstrated improvements in glycemic control, blood lipids and body weight. In some cases, this has been achieved to a greater degree than conventional dietary interventions [28]. Proposed mechanisms for this have been attributed to increased consumption of plant foods (naturally rich in minerals, vitamins, and antioxidants) and reduced intake of processed and red meats [29,30,31]. A study conducted to determine the nutritional adequacy of a low-fat vegan diet concluded that vegan diets have positive impacts on energy and plasma lipids [32]. The lipid lowering effects of plant-based diets can be attributed to negligible dietary cholesterol intake, reduced saturated fat content and cholesterol-lowering effects of soluble fiber [33]. These effects are important, as cardiovascular complications are one of the leading causes of worldwide morbidity and mortality in patients with T2DM.

A recently published systematic review and meta-analysis [34] confirmed that the consumption of vegan diets yields favorable results in some cardiometabolic health measures in overweight patients and those with T2DM. Despite there being an overlap in review content, our review is unique for several reasons: the most important being the current focus on patients with diabetes and the inclusion of VLCD diets. We included a side-by-side analysis by looking at both vegan and VLCD diets, being current popular approaches. Our inclusion criteria did not include individuals with pre-diabetes, but included studies of any length of intervention period, to facilitate a broader analysis for clinical relevance. This approach means that these results are translatable for patients, clinicians, and healthcare workers with an interest in dietary approaches to manage with type 2 diabetes and body weight control.

Our study assessed the evidence available to support very low-calorie diets (VLCD) and vegan diets for management of body weight and/glycemic control in patients with T2DM.

## 2. Materials and Methods

### 2.1. Registration of Review Protocol

A protocol was developed consistent with Preferred Reporting Items for Systematic Review and Meta-Analysis Protocols (PRISMA-P) and registered at International Prospective Register of Systematic Reviews. PROSPERO ID: CRD42022310299.

### 2.2. Databases and Search Terms

An electronic search of articles was conducted using Cochrane Library, MEDLINE (Ovid) and Embase on 15 February 2022. The following headings were included in the search strategy and used in all the fields or in combination as Medical Subject Headings terms: ‘Diabetes Mellitus, Type 2’ ‘type 2 diabetes’, ‘caloric restriction’, ‘VLCD’, ‘very low calorie diet’, ‘very low energy diet’, ‘semistarvation diet’, ‘crash diet’, ‘low fat vegan diet’, ‘LFVD’, ‘low fat vegetarian diet’, ‘VLED’, ‘plant based diet’, ‘diet, vegetarian’, ‘diet, vegan’, ‘English language’, ‘clinical trial’, ‘observational study’, and ‘randomized control trial’. Reference lists of previous systematic reviews or relevant original research articles were also searched to find studies that were not discovered in the initial database search. There were no restrictions placed regarding sex, ethnicity, race, sample size, publication status, or date of publication. Plant based diet and low-fat vegan diet will be used interchangeably.

### 2.3. Inclusion and Exclusion Criteria

RCTs, before-after studies, single-arm intervention trials, and non-randomized controlled trials were included if: (i) weight changes were included in the form of body mass index (BMI) (kg/m^2^) and/or weight (kg), (ii) fasting glucose/HbA1c was reported and (iii) people with type 2 diabetes were part of a studied population. Where available, secondary outcomes, i.e., LDL cholesterol, HDL cholesterol, total cholesterol, triglycerides, systolic blood pressure, diastolic blood pressure, urinary albumin, waist circumference, hip circumference, and waist-to-hip ratio were also analyzed. Eligible study populations were restricted to individuals with Type 2 Diabetes Mellitus.

Following database searches, results were imported to Rayyan, version 5:201 [35]. Eligible full-text papers were independently and critically assessed by two authors. A flowchart following the PRISMA statement was created to demonstrate the different phases of this process (Figure 1).

### 2.4. Data Extraction

A standardized form was used to extract data from the included studies to assist in assessment of study quality and evidence synthesis. Extracted information included: general information, study characteristics (sample size, country and year of publication, duration of follow up, intervention duration, intervention details, details about control groups or interventions not under review), participant characteristics (mean age, number of males and females), primary and secondary outcomes, results and conclusions, as well as information required for assessment of risk of bias.

### 2.5. Risk of Bias

The included studies were assessed using three tools for bias; the Cochrane RoB 2 tool for randomized control trials (RCT), the Cochrane ROBINS-I tool for non-randomized controlled trials (NRCT), and the National Institute of Health (NIH) quality assessment tool for other studies. RCT’s were assessed for bias across five domains, namely bias arising from the randomization process, bias due to deviations from intended intervention, bias due to missing outcome data, bias in measurement of the outcome, and bias in selection of the reported result. NRCT’s were assessed for bias across 7 domains, namely confounding, selection of participants, classification of interventions, deviation from intended interventions, missing data, measurement of outcomes, and selection of the reported result. The remaining studies were appraised using 12 questions to assess their internal validity.

### 2.6. Outcomes

The primary outcomes were body weight, measured in kilograms and glycemic control measured with HbA1c and fasting glucose. Secondary outcomes were fasting insulin, BMI, triglycerides, total cholesterol, HDL and LDL cholesterol, waist circumference, hip circumference, waist-to-hip ratio, systolic and diastolic blood pressure, and urinary albumin.

### 2.7. Intervention Groups

The intervention groups were chosen as common diets that patients ask about in clinic to offer health benefits or weight control.

VLCD Control: Non-dietary interventions such as behavioral therapy or usual medical care.

VLCD: Dietary intervention of less than 800 calories per day.

Vegan Control: Conventional diabetes diets recommended by National guidelines [36,37].

Vegan: Diet comprising vegetables, nuts and grains and excluding all animal products. The reported studies included in this review that studied VLCD’s as dietary interventions used non-dietary interventions as controls. There were insufficient papers that directly compare VLCD and vegan diets within in the same paper.

### 2.8. Data Synthesis and Analysis

Primary and secondary outcomes were analyzed with a final outcome assessment for each group; where appropriate, the baseline and final measurements within the study period were taken, e.g., in a study with baseline week 1, week 4, and week 8 measurements, only baseline and week 8 were compared [20]. Measurements were not taken from the end of the follow up period if the trial had one; the final data point used was that taken at the end of the intervention period. Studies that were deemed to be suitably homogeneous in terms of population and diets were pooled using a mean difference for studies of consistent scales (e.g., weight in kg, BMI in kg/m^2^) and fitted using a random effects meta-analysis. The mean difference between the intervention and comparator group is stated alongside 95% confidence intervals (95% CI), *p*-values and I2 statistics to assess statistical heterogeneity [38]. Statistical significance was determined to be reached at a *p* value of ≤0.05.

All sections of the results where forest plots have been created used data pooled solely from RCT’s and so cause-effect inferences can be drawn from these data. In instances where no forest plot has been generated and results are simply described, non -randomized experiments may have been included in the narrative, and so cause-effect relationships cannot be inferred. These instances are clearly outlined in the manuscript to reflect this circumstance.

### 2.9. Subgroups Used to Explain Heterogeneity

One comparison was investigating vegan diets against control diets, i.e., American Diabetes Association (ADA) diet and Korean Diabetes Association (KDA) diet, and the second comparison was between VLCD and non-dietary therapies, i.e., behavioral interventions. Heterogeneity, as demonstrated through utilization of these subgroups, explains why not all studies were suitable for pooled meta-analysis in each comparison. Analysis was also based on diet duration and adherence.

## 3. Results

### 3.1. Database Search

The search identified 3370 results. After exclusions, 16 studies were included for analysis [9, randomized control trials (RCT); 3, before-after studies (BAS); 1, non-randomized control trial (NRCT); 1, non-randomized pilot study (NRPS); 2, single arm intervention trials (SAIT)]. Additional publications were not found when reference lists were searched.

### 3.2. Study Characteristics

Details of the study characteristics for the 16 included studies can be seen in Appendix A. The 16 studies included 834 participants with type 2 diabetes. The mean ages ranged from 42.1 to 61.0 years. One study recruited male participants only [38], and another did not report whether they had both male and female participants or one gender exclusively [39]. Intervention and follow-up periods between studies varied; the shortest intervention and follow-up period was 4 days [39] and the longest intervention and follow-up period was 74 weeks [40]. Median intervention period was 17 weeks.

### 3.3. Risk of Bias

Of the nine RCTs, one study was deemed to be high risk of bias, five studies were moderate risk of bias, and three were low risk of bias. Nicholson et al. [41] was deemed to be high risk due to baseline differences between groups (Figure 2) and deviations from the intended intervention due to the trial context. One NRCT was of serious risk (Figure 3), five studies were of fair quality, and one was of good quality (Table 1).

### 3.4. Comparison of Vegan Diets vs. Control Diets

Results from the five studies investigating vegan diets found no significant reductions in bodyweight, improvement in anthropometric markers or improvement in glycaemic control, compared with the control diets (conventional diabetes diets) [28,40,41,46,47]. There was a highly significant reduction in LDL cholesterol linked to the vegan diet; total or LDL cholesterol was unaffected by dietary approach. Triglycerides were also unaffected by the diet approach; however, it is noted that the triglyceride levels were higher among patients in the vegan group.

#### 3.4.1. Primary Outcomes

##### Body Weight

Body weight was reported in four of the five studies (*n* = 249 participants) [28,40,41,47]. The pooled mean difference (MD) was −0.14 kg (95% CI −0.39, 0.11; *p* = 0.27; I^2^ = 0%; Figure 4). Standard mean difference favored no reduction in weight between vegan and control diets.

##### Fasting Glucose and HbA1c

Fasting glucose levels and HbA1c results were reported in four studies (*n* = 302 participants) [28,40,41,46]. The pooled MD was −0.01 mmol/L (95% CI −0.23, 0.22; *p* = 0.95; I^2^ = 0%; Figure 5) and −0.15% (95% CI −0.37, 0.08; *p* = 0.20; I^2^ = 0%; Figure 6) for glucose and HbA1c, respectively. Standard mean difference suggested no difference in fasting glucose levels between either diet, but rather a reduction in HbA1c after the vegan diets.

#### 3.4.2. Secondary Outcomes

##### Body Mass Index (BMI)

BMI results were reported in four studies (*n* = 331 participants) [9,28,40,41]. The pooled MD was −0.11 kg/m^2^ (95% CI −0.39, 0.18; *p* = 0.46; I^2^ = 39%; Appendix A). Standard mean difference suggested no difference in BMI between either diet. Appendix A is available in the Appendix A.

##### Triglycerides

Triglyceride results were reported in five studies (*n* = 342 participants) [28,39,40,41,42]. The pooled MD was 0.32 mmol/L (95% CI −0.15, 0.78; *p* = 0.18; I^2^ = 74%; Appendix A). Standard mean difference suggested no difference in triglycerides between either diet. Appendix A is available in the Appendix A.

##### Waist and Hip Circumferences and Waist-To-Hip Ratio

Waist circumference was measured and reported in three studies (*n* = 291 participants) [28,39,41]. The pooled MD was −0.16 cm (95% CI −0.39, 0.07; *p* = 0.17; I^2^ = 0%; Appendix A). Standard mean difference suggested no difference in waist circumference between either diet. Appendix A is available in the Appendix A.

Hip circumference was measured and reported in two studies [28,39]. Both papers showed statistically significant reductions in hip circumference when compared to baseline. Barnard et al. [28] noted a reduction of 3.9 cm (*p* < 0.001) after consumption of the vegan diet compared to 3.8 cm (*p* < 0.001) after the control diet. Barnard et al. [39] noted a loss of 3.4 cm (*p* < 0.001) after the vegan diet compared to 2.3 cm (*p* < 0.01) after the control diet.

Waist-to-hip ratio was measured and reported in two studies. Barnard et al. in 2006 [28] noted a loss of 0.02 cm (*p* < 0.01) after the vegan diet and 0.01 cm after the control diet. Barnard et al. in 2009 [39] noted a loss of 0.01 cm after the vegan diet and no reduction after the control diet.

##### Total Cholesterol

Total cholesterol was measured and reported in four studies (*n* = 249 participants) [28,39,40,42]. The pooled MD was −0.28 (95% CI −0.66, 0.09; *p* = 0.14; I^2^ = 45%; Appendix A). Standard mean difference suggested no difference in total cholesterol between either diet. Appendix A is available in the Appendix A.

##### HDL Cholesterol

HDL cholesterol levels were reported in all five studies (*n* = 342 participants) [28,38,39,40,41,42]. The pooled MD was 0.01 mmol/L (95% CI −0.20, 0.22; *p* = 0.93; I^2^ = 0%; Appendix A). Standard mean difference suggested no difference in mean HDL cholesterol between either diet. Appendix A is available in the Appendix A.

##### LDL Cholesterol

LDL cholesterol was reported in four out of the five studies (*n* = 331 participants) [28,40,46,47]. The pooled MD was −0.38 mmol/L (95% CI −0.60, −0.16; *p* < 0.001; I^2^ = 0%; Figure 7). Standard mean difference favored a vegan diet and suggested a significant decrease in mean LDL cholesterol compared with control diets.

### 3.5. Comparison of VLCD vs. Control

The results from all VLCD studies found no significant reduction in bodyweight or anthropometric markers compared with control. Key findings were the impact on glycemic control; both HbA1c and fasting glucose were significantly reduced by following the VLCD diet. A trend towards weight reduction and improvement of BMI was suggested by the data analyzed, also linked to the VLCD diet. Snel et al. [48] found the VLCD and exercise group to show a greater improvement in BMI compared to VLCD alone.

#### 3.5.1. Primary Outcomes

##### Body Weight

Body weight was reported in three studies (*n* = 227 participants) [22,49,50]. The pooled MD was −0.33 kg (95% CI −0.68, 0.03; *p* = 0.07; I^2^ = 32%; Figure 8). Standard mean difference suggested no difference between VLCDs and non-dietary therapies in reducing weight.

##### Fasting Glucose

Fasting glucose was reported in nine studies (*n* = 233 participants) [22,50]. The pooled MD was −1.51 mmol/L (95% CI −2.89, −0.13; *p* = 0.03; I^2^ = 86%; Figure 9). Standard mean difference favored VLCDs in reducing fasting glucose.

##### HbA1c

HbA1c was reported in six out of the eleven studies. Three studies were eligible for pooled meta-analysis (*n* = 297 participants) [22,49,50]. The pooled MD was −0.66% (95% CI −1.28, −0.03; *p* = 0.04; I^2^ = 74%; Figure 10). Standard mean difference favored a reduction in body weight after the VLCD diets. Caution should be taken when interpreting these results, as I^2^ is high. Standard mean difference favored VLCDs in reducing HBA1c.

#### 3.5.2. Secondary Outcomes

##### Body Mass Index (BMI)

BMI was reported in nine studies; however, the data could not be pooled, as a number of different interventions were investigated. They were not deemed to be clinically homogeneous for this review based on contextual differences.

##### Triglycerides

Triglycerides were reported in three studies (*n* = 227 participants) [22,46,48]. The pooled MD was 0.18 mmol/L (95% CI −0.61, 0.97; *p* = 0.65; I^2^ = 84%). Standard mean difference suggested no significant difference in mean triglyceride levels between intervention and control. However, the large heterogeneity may have been due to the study by Wing et al. [22] who offered a financial incentive (they asked their participants to deposit money before the intervention which would be returned if they completed homework monitoring adherence). There is a chance that the behavioral therapy group engaged in this to a greater degree. When that study was excluded, standard mean difference was −0.19 mmol/L (95% CI −0.93, 0.54; *p* = 0.61; I^2^ = 78%; Appendix A). Standard mean difference suggested no difference between VLCDs and non-dietary therapies in reducing triglycerides. Appendix A is available in the Appendix A.

##### Waist and Hip Circumferences and Waist-To-Hip Ratio

Waist circumference was reported in five studies (*n* = 220 participants) [16,20,46,47,49]. Steven and Taylor [20] reported a reduction of 12 cm and 12.4 cm for their short intervention duration (SD) and long intervention duration (LD) groups, respectively (no statistical significance). Lim et al. [16] reported a decrease of 13.2 cm (*p* < 0.05) in the intervention group. Jazet et al. [49] reported a reduction of 19.1 cm (*p* < 0.001). Taheri et al. [46] reported a reduction of 11.44 cm. Snel et al. [47] reported significant decreases (*p* = 0.049) in both VLCD and VLCD and exercise groups of 19 cm and 25 cm, respectively. In summary, waist circumference decreased from the baseline across all studies.

Hip circumference (HC) was reported in two studies (*n* = 49 participants) [16,20]. Both studies found a reduction in hip circumference; reported a 10 cm decrease (*p* < 0.05) and reported a reduction of 7.8 cm and 7.4 cm for the short and long duration groups, respectively. Both studies showed reduction in HC albeit the latter did not show statistically significant results.

Waist-to-hip ratio was reported in three studies (*n* = 196 participants) [16,20,46] and a 0.1 cm decrease was reported by [16], while [20] reported a 0.03 cm decrease, and [46] reported a reduction of 0.04 cm and 0.05 cm for the short and long duration groups, respectively (no statistical significance).

It is important to note that Steven and Taylor [20], Lim [16], and Jazet et al.’s [49] studies were non-randomized, and so the results from these papers in this section cannot be used to draw cause-effect inferences.

##### Total Cholesterol

Total cholesterol was reported in ten studies [22,28,39,40,41,42,43,45,48,51]. Three of these papers were suitable for pooled meta-analysis (*n* = 227 participants) [22,46,48]. The pooled MD was 0.37 mmol/L (95% CI −0.33, 1.07; *p* = 0.30; I^2^ = 80%; Appendix A). Standard mean difference suggested no difference between VLCDs and non-dietary therapies in reducing total cholesterol. Appendix A is available in the Appendix A. The large heterogeneity was due to the study by Taheri et al. [46], which used a different type of non-dietary therapy to that employed by the other two studies. After that study was excluded, the standard mean difference was 0.01 mmol/L (95% CI −0.44, 0.47; *p* = 0.95; I^2^ = 0%) and favored a reduction in total cholesterol after non-dietary therapies.

The remaining seven studies were not meta-analyzed. Steven and Taylor [20] reported an improvement of 1.0 mmol/L (*p* = 0.004) and 1.1 mmol/L (*p* = 0.001) in the short duration and long duration groups, respectively. Lim et al. [16] reported a 0.8 mmol/L improvement (*p* < 0.005). Jackness et al. [38] noted a 0.7 mmol/L reduction (*p* < 0.001) in the VLCD group compared to 0.98 mmol/L (*p* < 0.01) in the Roux-en-y gastric bypass (RYGB) group. Snel et al. [47] reported a 0.6 mmol/L decrease in the VLCD group compared to 0.9 mmol/L in the VLCD and exercise group (no statistical significance). Snel et al. [43] reported a 0.7 mmol/L improvement. Skrha et al. [45] reported a 0.5 mmol/L decrease (*p* < 0.001) compared to 1.0 mmol/L (*p* < 0.01) in the control group. Statistically significant reductions from the baseline were observed across all studies apart from Snel et al. [47].

It is important to note that Steven and Taylor [20], Lim [16], Jackness et al. [38], Snel et al. [43], and Skrha et al.’s [45] studies were non-randomized, and so the results from these papers in this section cannot be used to draw cause-effect inferences.

##### Fasting Insulin

Fasting insulin was reported in seven out of the eleven studies. Two of those studies were suitable for pooled meta-analysis (*n* = 80 participants) [22,48]. The pooled MD was −0.12 pmol/l (95% CI −0.57, 0.33; *p* = 0.61; I^2^ = 0; Appendix A). Standard mean difference suggested no difference in mean fasting insulin after VLCDs or non-dietary therapies. Appendix A is available in the Appendix A.

##### HDL Cholesterol

HDL cholesterol was reported in ten out of the eleven studies. Three of those studies were suitable for pooled meta-analysis (*n* = 227 participants) [22,46,48]. The pooled MD was 0.11 mmol/L (95% CI −0.59, 0.80; *p* = 0.76; I^2^ = 80%; Appendix A). Standard mean difference suggested no difference in mean HDL cholesterol after VLCDs or non-dietary therapies. Appendix A is available in the Appendix A.

##### LDL Cholesterol

LDL cholesterol was reported in seven out of the eleven studies. Two of those studies were suitable for pooled meta-analysis (*n* = 194 participants) [49,50]. The pooled MD was 0.59 mmol/L (95% CI −0.18, 1.35; *p* = 0.13; I^2^ = 80%; Figure 11). Standard mean difference favored a greater reduction in LDL cholesterol after non-dietary therapies.

##### Systolic Blood Pressure

Systolic blood pressure was reported in three out of the eleven studies [20,46,47]. Only one study was eligible for meta-analysis and so a forest plot was not created. Taheri et al. [46] reported mean changes of −8.19 mmHg and −4.42 mmHg in the intervention and control groups, respectively but this difference was not significant.

The other two studies did report a significant decrease in mean systolic blood pressure [20,47]. Steven and Taylor [20] reported changes of −19 mmHg (*p* < 0.003) and −27 mmHg (*p* < 0.001) in the short duration (SD) and long duration (LD) groups, respectively, and Snel et el. [47] reported changes of −21 mmHg in the VLCD only group and −13 mmHg in the VLCD + exercise group (both *p* < 0.05). These results across all papers show that there is an overall trend of a greater reduction in mean systolic blood pressure in participants following a VLCD compared to those following non-dietary therapies. It is important to note that Steven and Taylor’s [20] study was non-randomized, and so the results from this paper in this section cannot be used to draw cause-effect inferences.

##### Diastolic Blood Pressure

Diastolic blood pressure was reported in three out of the eleven studies [20,46,47]. Only one study was eligible for meta-analysis and so a forest plot was not created. Taheri et al. [46] reported mean changes of −5.60 mmHg and −2.24 mmHg in the intervention and control groups, respectively but this difference was not significant. Steven and Taylor [20] reported changes of −9 mmHg (*p* < 0.007) and −10 mmHg (*p* < 0.003) in the SD and LD groups, respectively, and Snel et al. [47] reported changes of −9 mmHg in the VLCD only group and −6 mmHg in the VLCD + exercise group (both *p* < 0.05). These results across all papers show that there is an overall trend of a greater reduction in mean diastolic blood pressure in participants following a VLCD compared to those following non-dietary therapies.

It is important to note that Steven and Taylor’s [20] study was non-randomized, and so the results from this paper in this section cannot be used to draw cause-effect inferences.

##### Urinary Albumin

None of the eleven studies examining VLCDs reported urinary albumin.

## 4. Discussion

### 4.1. Summary of Main Findings

This study included 16 studies with 834 participants with varied risk of bias and found that both vegan diets and VLCD can offer some clinical improvement in people with type 2 diabetes. Studies have shown that a low-fat vegan diet can drastically lower total cholesterol and LDL cholesterol which in combination reduces the risk of cardio-vascular disease. This is significant as cardiovascular disease is one of the primary comorbidities of T2DM and a prominent cause of early mortality in populations with diabetes [33]. Most health parameters showed clinically significant but not statistically significant improvements when following a vegan diet compared to a conventional diabetes diet. Exceptions to this include fasting glucose, HDL cholesterol, and systolic and diastolic blood pressure, which showed no notable difference. A modest increase in triglycerides was also reported in vegan participants of Barnard et al.’s 2018 study [47]. This is consistent with previous studies which have found when compared to omnivorous diets, vegan and vegetarian diet groups have higher triglyceride levels in clinical trials and lower levels in observational studies. This may be due to a higher carbohydrate intake in vegan diets which has been found to raise triglyceride levels [51]. Typically, this is not to a degree that is statistically or clinically significant.

Vegan diets have many advantages including the lack of restriction on caloric intake, absence of portion size calculations and the simplicity of understanding the diet (elimination of animal products) [52]. This was highlighted in the study by Nicholson et al. [41] where it was reported that participants following the vegan diet lost 7.2 kg of weight with no restrictions placed on energy intake. In comparison to conventional diabetes diets which are centered around the restriction of calories and strict portion control, vegetarian and vegan diets have been reported to be easier to follow in combination with exercise due to suppression of hunger signals [53]. As both diet and exercise are two key components for the effective management of T2DM, this would be an important factor to consider for T2DM patients.

There are several possible mechanisms which may help in explaining the favorable outcomes observed with plant-based diets. Vegan diets generally emphasize the consumption of fruits, vegetables, legumes, nuts, and wholegrains which are abundant in vitamins and minerals, fiber, antioxidants, unsaturated fatty acids, and phenolic compounds. Studies have shown that these foods can lower weight gain long term and ameliorate systemic inflammatory pathways involved in the disease processes of type 2 diabetes [53,54]. These diets also discourage or eliminate processed and red meats which are known to adversely affect health parameters in type 2 diabetes, possibly as a result of high levels of heme iron and dietary cholesterol [55].

Despite the high carbohydrate content of a vegan diet, all trials reviewed demonstrated glucose lowering effects with more pronounced changes seen in participants adopting a conventional hypocaloric diet. This may be attributable to the higher fiber content; dietary fiber reduces the postprandial response of glucose by processes, such as reduced gastric emptying and subsequent slowing of starch digestion and the glucose absorption. In addition, glucagon-like peptide 1 (GLP-1) plays a key role; besides slowing gastric emptying, glucose uptake and disposal is improved in peripheral tissues, especially those which are insulin-dependent. GLP-1 reduces the production of glucose in the liver through inhibiting glucagon secretion [56]. Some prospective cohort studies suggest that the fiber intake from cereal is what reduces the long-term risk of T2DM rather than fruit and vegetable fiber [57,58].

The lower fat content of a vegan diet could also contribute to improvement in cardiometabolic risk factors. Diets rich in fat increase intramyocellular lipid (IMCL) concentration through downregulation in skeletal muscle of mitochondrial oxidative phosphorylation genes [59]. Excess IMCL has cytotoxic effects on the mitochondria through overproduction of ROS and metabolic stress thereby promoting insulin resistance [60]. A study by Goff et al. [61] comparing vegans and omnivores found a significantly lower concentration of IMCLs in the soleus muscle of the vegan cohort. The quality of the fat source is also an important factor to consider as it has been found that the association between IMCLs and insulin resistance is not a definitive cause-effect relationship.

Akin to the effects of vegan diets, VLCD’s were found to offer clinical improvements. The effect on glycemic control was marked; both fasting glucose and HbA1c were significantly reduced from baseline and when compared with non-dietary therapies.

Time of follow up was important however, with Wing et al. [22] finding that although weight loss was observed over the 8-week study period, long term weight loss (1 year) was not significant. This was also reported by Wadden and Stunkard [62]. Both studies found that loss was greater in the VLCD participants and so too was weight gain over the year of follow-up. Thus, it was concluded that VLCD diets provided no benefits regarding weight loss in the long term compared to a non-dietary therapy. Snel et al. [48] however found that weight loss was improved to a greater degree when combined with exercise. They established that in the long term (18 months following study), exercise was key in maintenance of diet-based improvements in weight loss.

The mechanism behind VLCD’s efficacy is simply that of any diet involving caloric restriction; being in a calorie deficit results in the body metabolizing fat leading to rapid weight loss and improvement in anthropometric markers. Interestingly, in the context of glycemic control, recent evidence suggests that with this weight loss also comes concurrent improvement of pancreatic beta cell function and subsequent improved insulin sensitivity [17,18]. The results of this meta-analysis suggest a significant improvement in glycemic control in people with type 2 diabetes who adhere to a VLCD in contrast to non-dietary therapies; both mean fasting glucose levels and mean HbA1c significantly decreased in the intervention groups. This evidence is corroborated by a recent review and meta-analysis by Sellehewa et al., [63], who included 17 studies looking at VLCD’s in people with type 2 diabetes. The mean HbA1c reduction reported was 1.4%, ranging from 0.1% to 3.1% across the various studies.

Advantages of a VLCD are that they promote very rapid weight loss and thus can provide clinical efficacy in a short period of time, as well as often coming in liquid meal replacement form making it easier to ensure adequate vitamin, mineral, and macronutrient intake. Moreover, VLCD’s have the additional benefit of equally rapid improvement in concomitant medical issues, most notably cardiovascular risk factors. This was noted in our study which found that systolic and diastolic blood pressure were significantly reduced in VLCD groups compared with non-dietary therapy groups. There is a cornucopia of evidence proving the benefits of blood pressure reduction in a whole variety of cardiovascular diseases; a 2016 meta-analysis by the National Institute for Health Research [64] found that reducing systolic blood pressure by only 10 mmHg reduced the risk of major cardiovascular events by 20%, stroke by 27%, heart failure by 28%, coronary heart disease by 17%, and deaths from all causes by 13%.

### 4.2. Strengths and Limitations

Our study has several key strengths. Firstly, it provides a targeted insight on dietary approaches for patients with type 2 diabetes, which considers the most recent weight loss diets, acknowledging the shift toward plant-based eating. This information is useful for clinicians and healthcare practitioners to share with patients. This review added to previous similar reviews [34] by incorporating side-by-side analysis by looking at both vegan and VLCD diets as well as excluding focusing solely on individuals with T2DM and including studies with any length of intervention periods to carry out a broader analysis for clinical translation. We also followed guidance by utilizing double data extraction and robustly synthesized our results which lends further overall strength to the evidence presented. Our study has several limitations, primarily the included study designs had a degree of heterogeneity, possibly explained by culture. Additionally, studies were small and lacked power. A lack of standardization, for example in variation between studies as to what was classified as a non-dietary therapy was an issue. Adherence was often self-reported, and thus lower adherence levels may play a part in the lack of significant differences in clinical outcomes, particularly for vegan diets. The final limitation was the duration of intervention and follow up; many of our included studies had very short intervention periods with no long term follow up making it difficult to translate results into a real-world clinical setting with potential for implementation as part of national diabetes guidelines. Sufficient evidence to suggest that these diets have sustainable beneficial effects is therefore crucial.

## 5. Conclusions

The systematic review and meta-analysis have shown that both low-fat vegan diets and very low-calorie diets are more effective than conventional diabetes diets and non-dietary therapies for reducing LDL cholesterol and inducing good glycemic control, respectively, in patients with T2DM. There is scope for these diets to be integrated into national guidelines and recommendations to a much greater extent, perhaps even as first line choices when individually suitable. For this to come to fruition, larger scale RCT’s with longer intervention periods and thoroughly analyzed follow-ups with in-depth analysis of attrition and adherence would be recommended, in order to solidify the evidence that vegan and very low-calorie diets can be maintained to an appropriate degree to have a lasting favorable impact in type 2 diabetes outcomes.

## Figures and Tables

**Figure 1 nutrients-14-04870-f001:**
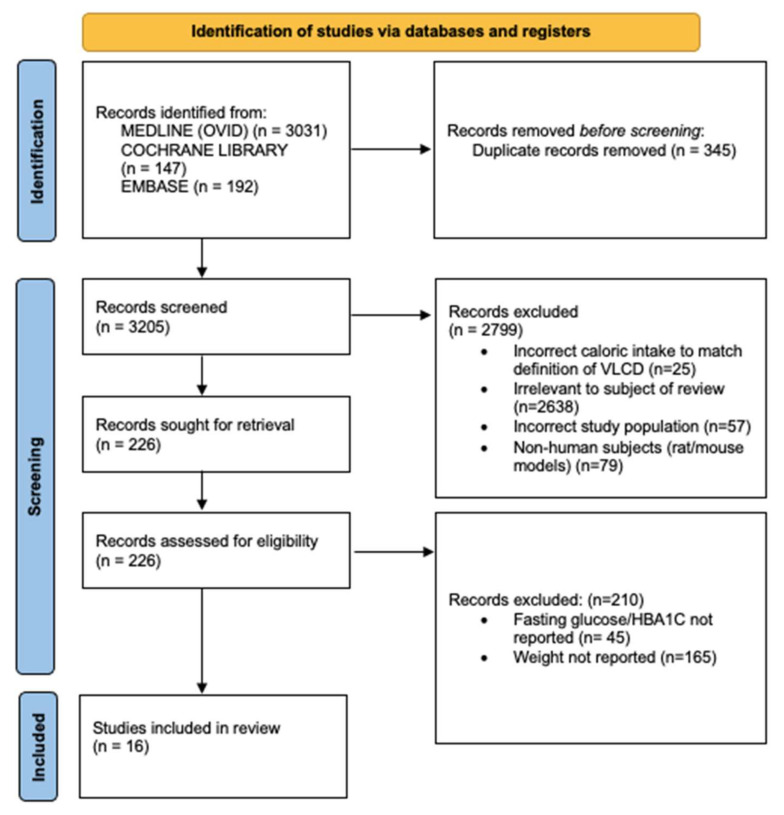
PRISMA illustrating the study selection process.

**Figure 2 nutrients-14-04870-f002:**
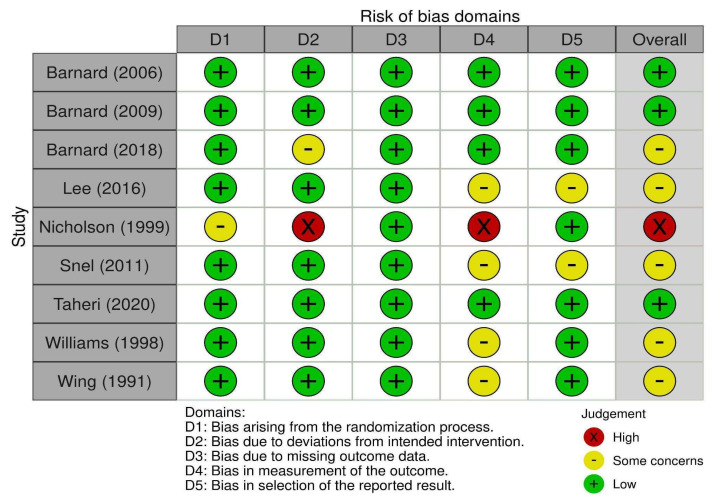
Risk of bias presented using Cochrane RoB 2 tool for the randomized controlled trials [22,28,40,41,43,46,47,49,50].

**Figure 3 nutrients-14-04870-f003:**
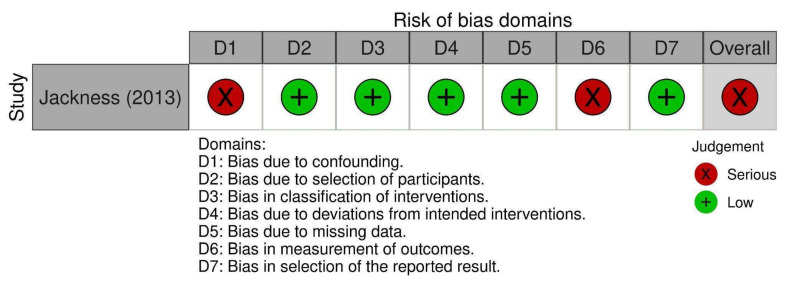
Risk of bias presented using Cochrane ROBINS-I tool for the non-randomized control trial [42].

**Figure 4 nutrients-14-04870-f004:**
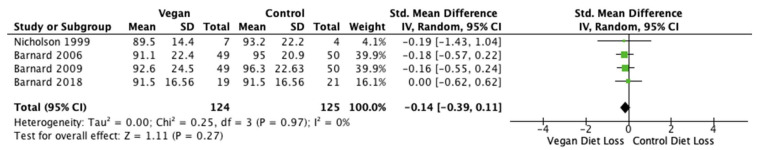
Forest plot comparing weight loss after consumption of low-fat vegan and control diets [28,40,41,47].

**Figure 5 nutrients-14-04870-f005:**
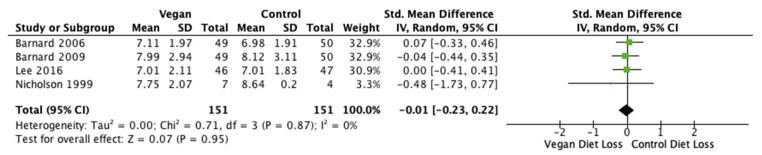
Forest plot comparing mean fasting glucose levels after the consumption of low-fat vegan and control diets [28,40,41,46].

**Figure 6 nutrients-14-04870-f006:**
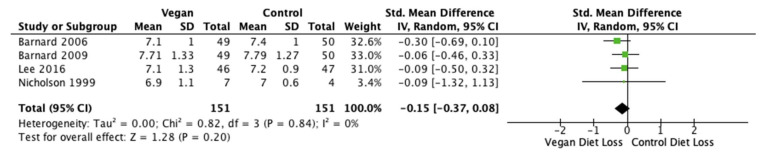
Forest plot comparing mean HbA1c levels after consumption of low-fat vegan and control diets [28,40,41,46].

**Figure 7 nutrients-14-04870-f007:**
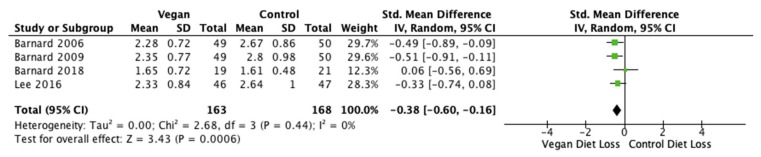
Forest plot showing mean LDL cholesterol after consumption of low-fat vegan and control diets [28,40,46,47].

**Figure 8 nutrients-14-04870-f008:**
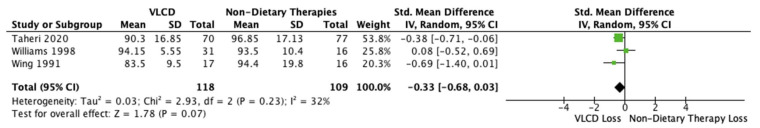
Forest plot comparing the reduction in body weight between VLCD and non-dietary therapies [22,49,50].

**Figure 9 nutrients-14-04870-f009:**
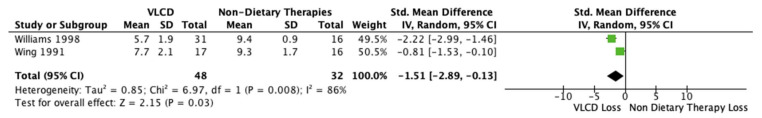
Forest plot comparing fasting glucose levels after VLCD and non-dietary therapies [50,22].

**Figure 10 nutrients-14-04870-f010:**
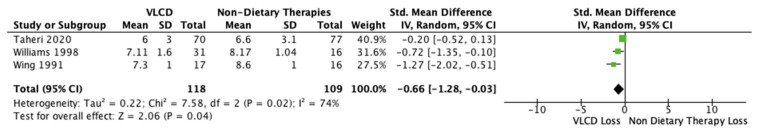
Forest plot comparing HbA1c levels after VLCD and non-dietary therapies [22,49,50].

**Figure 11 nutrients-14-04870-f011:**
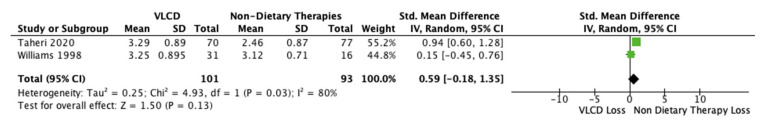
Forest plot showing mean LDL cholesterol after VLCD and non-dietary therapies [46,50].

**Table 1 nutrients-14-04870-t001:** Risk of bias presented using the NIH Quality Assessment Tool for non-randomized trials (i.e., before-after studies, non-randomized pilot study, and single arm intervention trials).

Question		Lim (2011) [16]	Snel (2012) [43]	Jazet (2008) [44]	Skrha (2005) [45]	Steven & Taylor (2015) [20]	Teeuwisse (2012) [39]
1	Was the study question or objective clearly stated?	√	√	√	√	√	√
2	Were the eligibility/selection criteria for the study population prespecified and clearly described?	√	√	√	√	√	√
3	Were the participants in the study representative of those who would be eligible for the test/service/intervention in the general or clinical population of interest?	√	√	√	√	√	√
4	Were all eligible participants that met the prespecified entry criteria enrolled?	NR	NR	NR	NR	NR	NR
5	Was the sample size sufficiently large to provide confidence in the findings?	√	√	√	✕	✕	√
6	Was the test/service/intervention clearly described and delivered consistently across the study population?	√	√	√	√	√	√
7	Were the outcome measures prespecified, clearly defined, valid, reliable, and assessed consistently across all study participants?	√	√	√	√	√	√
8	Were the people assessing the outcomes blinded to the participants’ exposures/interventions?	√	NR	NR	NR	NR	NR
9	Was the loss to follow-up after baseline ≤ 20%? Were those lost to follow-up accounted for in the analysis?	√	O	√	√	√	√
10	Did the statistical methods examine changes in outcome measures form before to after the intervention? Were statistical tests done that provided *p* values for the pre-to-post changes?	√	√	√	√	√	√
11	Were outcome measures of interest taken multiple times before the intervention and multiple times after the intervention (i.e., did they use an interrupted time-series design)?	✕	✕	✕	✕	✕	✕
12	If the intervention was conducted at a group level (e.g., a whole hospital, a community, etc.) did the statistical analysis take into account the use of individual-level data to determine the effects at the group level?	NA	NA	NA	NA	NA	NA
Summary Quality ^1^		ii	i	i	i	i	i

^1^ Quality was rated as ‘0′ for poor (0–3 out of 12 questions); ‘i’ for fair (4–8 out of 12 questions); ‘ii’ for good (9–12 out of 12 questions). √, Yes; ✕, No; NA, not applicable; NR, not reported.

## Data Availability

Data described in the manuscript, code book, and analytic code will be made available upon re-quest.

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
