# Peer review of "Investigating the Effectiveness of Very Low-Calorie Diets and Low-Fat Vegan Diets on Weight and Glycemic Markers in Type 2 Diabetes Mellitus: A Systematic Review and Meta-Analysis"

_nutrients, 2022, doi:10.3390/nu14224870_

Round 1
Reviewer 1 Report
Thank you for the opportunity to review this paper. This study provide a systematic review and meta-analysis on the effectiveness of Very Low-Calorie Diets and low-Fat Vegan Diets on Weight and Glycemic Markers in Type 2 Diabetes Mellitus. The review summarises data from related clinical, non-clinical and non-dietary intervention studies. Analysis/findings in this study indicated that vegan diets and VLCD can be used as dietary guidelines to improving metabolic outcomes in subjects with type 2 diabetes. However, the organization and presentation of results/findings in this manuscript should be improved. There ae multiple points that require clarifications.
Abstract:
Line 24: Please change “16 studies with 834 individuals” to “Sixteen studies with a total of 834 individuals”.
Line 28: Please state MD in full at first mention. MD: Mean difference
Lines 25-32: While the authors stated that “Both diets reduced bodyweight and improved glycemic control from baseline”, it was later stated that significant differences were only observed in the VLCD diet group. Yet, the conclusion stated that “A clear association exists between following VLCDs and vegan diets and the improvement of anthropometric markers in patients with Type 2 Diabetes”. The authors need to clarify their findings and revise the conclusion.
Introduction:
Lines 66-68: The authors stated that “A study conducted to determine the nutritional adequacy of a low-fat vegan diet concluded that both vegan and conventional diabetes diets have positive impacts on energy and plasma lipids”, the conclusion that both diets have positive impacts on energy and plasma lipids are not addressing the main aim of the study which is to determine nutritional adequacy. What point is the author trying to convey? Is it that both diets have same nutrition or both diets have same impact on energy and plasma lipids? If the authors are trying to convey that both diets have impact on energy and plasma lipids, then the question would be so why not conventional diet instead of low-fat vegan diet?
Lines 58-72: The authors used plant-based diet and low-fat vegan diet interchangeably throughout the paragraph. It will be clearer to use one term (low-fat vegan diet as mentioned in manuscript title) or explain that they are the same.
Materials and methods:
Lines 102-105: The authors stated that “Included studies were: RCTs, before-after studies, single-arm intervention trials and non-randomized controlled trials were included if: (i) weight changes were included in the form of BMI (kg/m2 )/weight (kg), (ii) fasting glucose/HbA1c was reported and (iii) people with type 2 diabetes were a studied population”. Do the authors mean “RCTs, before-after studies, single-arm intervention trials and non-randomized controlled trials were included if: (i) weight changes were included in the form of BMI (kg/m2 )/weight (kg), (ii) fasting glucose/HbA1c was reported and (iii) people with type 2 diabetes were a studied population”? The “included studies were” seem unnecessary.
Line 105: Please change “people with type 2 diabetes were a studied population” to “people with type 2 diabetes were part of a studied population”.
Lines 105-106: Please change “Where available, secondary outcomes were also collected” to “Where available, secondary outcomes i.e., LDL cholesterol, HDL cholesterol, total cholesterol, triglycerides, systolic blood pressure, diastolic blood pressure, urinary albumin, waist circumference, hip circumference, and waist to hip ratio were also analysed”.
Lines 108-109: What do the authors mean by “eligible study populations included healthy adults”? This review should only be reviewing people with T2DM. Please clarify.
Figure 1: Please remove the “Reports not retrieved (n=0)” from the flowchart.
Figure 1: Please provide more description on why the 210 reports were excluded.
Line 137: Please remove “,”
Lines 142-148 (Section 2.7): What do the authors mean by different intervention groups? Why non-dietary interventions are included in this review and if they are non-dietary interventions, why label them as control diet? Should it just be control? What are the controls for VLCD?
Line 151: Please change “where appropriate the final measurement in a study period was taken” to “where appropriate, the baseline and final measurements within the study period were taken”
Line 151: Does the final measurement refer to measurement taken after intervention or after the whole study period (inclusive of follow-up where no diet intervention is provided)?
Line 162: Please change “second studied very low calorie diets with non-dietary therapy” to “second comparison was between VLCD and non-dietary therapies”
Lines 164-165: Please change “Diets were also analysed by duration and adherence” to “Analysis were also based on diet duration and adherence”.
Line 165: Was the adherence analysed during the follow-up period?
Line 168: Please insert “,” after “After exclusions”
Results: The presentation of the data is confusing and there are too many figures in the results section.
Supplementary Table 1: The version contains tracked changes that were not addressed. Please kindly clarify on the mean age column for studies by Snel et al. and Williams et al.
Lines 198-203: Although the authors stated that vegan diets are more effective than control diet, they should specify clearly if these differences are significant for which primary or secondary outcomes. From the figures, it seem only LDL cholesterol (secondary outcome) is significantly different between vegan and control diet groups.
Lines 202-203: The authors stated that “these differences were not statistically significant” but this sentence seems to only imply that the differences for HbA1c between vegan and control diet groups were not statistically significant. In this case, differences in all outcomes (primary and secondary) except LDL were not statistically significant. Please indicate clearly.
Figures 4-6: The figure legend should include description of the symbols used in the figures (dot, square and diamond)-the symbols should be the specific study that was reviewed. Please kindly indicate. Figures 4-6 can be combined together as Figure 4A/B/C.
Figures 7-15: The figure legend should include description of the symbols used in the figures (dot, square and diamond)-the symbols should be the specific study that was reviewed. Please kindly indicate. Figures 7-15 can be combined together as Figure 5A/B/C etc.
Tables 2, 3 and 5: Please include border lines so as to better indicate the study or subgroups for each outcome.
Table 4: This should be placed after Figure 18.
Figures 16-18: The figure legend should include description of the symbols used in the figures (dot, square and diamond)-the symbols should be the specific study that was reviewed. Please kindly indicate. Figures 16-18 can be combined together as Figure 6A/B/C.
Figures 19-23: The figure legend should include description of the symbols used in the figures (dot, square and diamond)-the symbols should be the specific study that was reviewed. Please kindly indicate. Figures 19-23 can be combined together as Figure 7A/B/C etc.
Lines 340-342: What do the authors mean by “BMI was reported in nine studies, however, the data couldn’t be pooled as a number of different interventions were investigated. This was not deemed to be clinically homogeneous for this review which focused on only two intervention”? If so why less than 9 studies were presented in Table 5?
Discussion:
Lines 454-457: Grammatical error, please clarify.
Lines 454-455: Could the lack of significant differences for vegan diet on clinical outcomes be attributed to the duration of the study intervention or non-adherence/compliance during follow-up period?
Lines 458-462: What could be the possible mechanism that vegan diet lead to higher (non-statistically significant) levels of triglycerides?
Lines 493-500: Please change “ICML” to “IMCL”.
Line 518: Please change “im-proved” to “improved”
Line 539: Please change “Our study did have limitations” to “Our study have several limitations”
Lines 539-547: Multiple grammatical errors, the points are unclear. Please clarify.
Line 544: Apart from follow-up period, how about intervention period?
Author Response
Reviewer 1 Thank you for your review and comments. All points are appreciated and addressed in the Table attached; these have improved the clarity of our paper.
Reviewer 2 Report
The authors have conducted an interesting review of the effects of low-calorie or -fat diets on diabetes. Please consider the following comments:
Comment 1: When providing the itemized response to comments, please indicate, for every comment, the specific lines where the modifications to the texts are located in the UPLOADED REVISED version of the manuscript. Thank you.
Comment 2: No specific criterion (P value) to define if an effect was statistically significant is presented in Materials and Methods. The authors can choose not to use the arbitrary criterion of 0.05 (or any other value) as a breakpoint to separate significant from non-significant effects, and that is okay. However, ambiguous terminology and contradictory ways to express ideas regarding effects and if they were or not statistically significant must be avoided. For example, the abstract says (lines 27-28) that "VLCDs REDUCED body weight... (p=0.07)... however, STATISTICAL SIGNIFICANCE was only observed...(p=0.03)." Also, regarding the same finding, line 319 says "FAVOURED a reduction in weight..." "Reduced" is emphatic; therefore refers a degree of confidence in the inference. For this, the p-value is not an issue (a specific value can be set arbitrarily where the authors prefer, or can be considered as a continuous value). However, the phrase "statistical significance was ONLY" contradicts the strength of the previous inference. Also, "favored" is a clear statement. Please be clear on the way p values are used to build inferences and be consistent in following the criteria in the way sentences are built up. Please carefully check the whole manuscript must be carefully checked for this criterion, and consider adapting the Discussion accordingly if necessary.
Comment 3: Please clarify the way the unit in line 104 is expressed for BMI.
Comment 4: Please check the whole manuscript for every abbreviation used to be defined the first time mentioned. e.g., BMI in lines 139 (defined) and 104 (not defined.
Comment 5: This is a major issue. Only 50% of the studies (9 of 16) in the review were actual randomized experiments. This data cannot support cause-effects inferences. They can be built up only based on those specific studies. Otherwise, all the manuscripts must be adapted to remove every statement inferring a cause-effect relationship. In cases where all studies were actual RCT (e.g., Table 2), please be explicit stating so in a caption under the table or figure. Readers should be able to find this specific piece of information without needing to go to supplementary material. Please find a way to clearly separate data and results providing cause-effect inferences from other that does not. Also, please include a note in the Discussion about this.
Comment 6: The references of the studies included in the forest plots must be placed next to every effect size; as standard forest plots.
Comment 7: Please ensure that the readers can easily find where every study was used in your review.
Author Response
Reviewer 2: Thank you for your review and comments. All points are appreciated and addressed in the Table attached. One point remains and we seek Editorial guidance on this. This relates to the style request to merge tables into panels or keep separate - thank you.

Round 2
Reviewer 1 Report
Thank you for the opportunity to review the revised manuscript. The authors have addressed all the comments except the request to combine multiple figures-pending editorial decision. However, in the reviewer’s opinion, there are too many figures in the manuscript. There are still parts in the manuscript that require revision and clarification.
Abstract:
Line 26: Please change “however LDL cholesterol was significantly decreased by vegan diet” to “However, LDL cholesterol was significantly decreased by vegan diet.”
Lines 30-32: Please change to “VLCD diet intervention is associated with improvement in glycaemia control in patients with Type 2 Diabetes”.
Introduction:
Lines 75-80: This section did not highlight clearly how this review is different from reference 34. The authors stated that most important being the current focus on diabetes per se but reference 34 also focus on diabetes (and obesity). While the authors mentioned that their review did not include individuals with pre-diabetes (unlike reference 34), the authors did not explain why is it important that their review only focuses on patients who are diagnosed with type 2 diabetes condition. This section can be better phrased to improve clarity.
Lines 81-83: The authors stated that “Our study assessed the evidence available to support very low-calorie diets (VLCD) and vegan diets for management of body weight and glycemic control exclusively in T2DM”. Did the authors only assessed studies that support these diets for body weight and glycemic control in T2DM or did the authors assessed studies that examined effect of these diets on body weight and glycemic control in T2DM? Please clarify.
Figure 1: It is confusing why the authors decided to put “Studies included in review (n=16)” and “Reports of included studies (n=16)”. Overall, only 16 studies were included so why is there a need for 2 different phrasing? Why do the authors use “records” and “reports” interchangeably? Please use either one of them consistently throughout the Figure. For records excluded (n=2799): please state the reasons/description in point form, for reports excluded (n=210): please state the reasons/description in point form.
Lines 157-163: Can the authors please clarify how the intervention groups were derived? Based on the reported studies? Does it mean that the reported studies included in this review which studied VLCD as dietary intervention have used non-dietary interventions as controls? Otherwise, please state why VLCD diet cannot be compared against vegan control or why vegan diet cannot be compared against non-dietary controls?
Line 161: Please include description or cite appropriate reference for the “Vegan control” Conventional diabetes diets recommended by National guidelines
Lines 162-163: Please change to “Vegan: Diet comprising vegetables, nuts and grains and excluding all animal products”
Lines 222-228: The authors did not mention LDL cholesterol which was significantly improved following vegan diet. For “The data did however suggest a trend towards reduction in weight and improvement of anthropometric markers”, the authors should state clearly that the reduction and improvement in these parameters are due to vegan diet. Lines 227-228 should be phrased in relation to the vegan diet and focus on triglycerides as the mean difference for fasting glucose is very small e.g. Triglyceride levels was higher among patients on vegan diet. Moreover, these data are not significant.
Lines 338-339: The authors should specify that the trend is observed following VLCD.
Lines 505-506: Please include citation for the sentence “This may be due to a higher carbohydrate intake in vegan diets which has been found to raise triglyceride levels”.
Line 583: Under this section which titled “strengths and limitations”, only limitations were stated but there were no strengths.